# RESMAX: AN ALTERNATIVE SOFT-GREEDY OPERATOR FOR REINFORCEMENT LEARNING

## ABSTRACT

Soft-greedy operators, namely $\varepsilon$-greedy and softmax, remain a common choice to induce a basic level of exploration for action-value methods in reinforcement learning. These operators, however, have a few critical limitations. In this work, we investigate a simple soft-greedy operator, which we call resmax, that takes actions proportionally to their suboptimality gap: the residual to the estimated maximal value. It is simple to use and ensures coverage of the state-space like $\varepsilon$-greedy, but focuses exploration more on potentially promising actions like softmax. Further, it does not concentrate probability as quickly as softmax, and so better avoids overemphasizing sub-optimal actions that appear high-valued during learning. Additionally, we prove it is a non-expansion for any fixed exploration hyperparameter, unlike the softmax policy which requires a state-action specific temperature to obtain a non-expansion (called mellowmax). We empirically validate that resmax is comparable to or outperforms $\varepsilon$-greedy and softmax across a variety of environments in tabular and deep RL.

## 1 INTRODUCTION

Every online decision-making problem involves the exploration-exploitation dilemma: to choose between taking actions that seem best in the short term and alternatives that can provide potentially valuable information about action selection for the long run. The trade-off between exploration and exploitation has been well studied in reinforcement learning (RL). The literature on exploration is vast, particularly in the field of RL theory. Many methods with important theoretical insights are not easy to use or are not efficient (Jaksch et al., 2010; Wang et al., 2020). More recent exploration methods developed for deep RL are more practical to use, such as those that use reward bonuses and counts (Bellemare et al., 2016; Taiga et al., 2020; Pathak et al., 2017). These developments, however, are as yet quite new and several studies have highlighted inconsistent performance and difficulties tuning hyperparameters (Taiga et al., 2020; Yasui et al., 2019).

Due to this, most work continues to use a basic form of exploration: soft greedy operators, such as softmax (Luce, 1959) and $\varepsilon$-greedy (Watkins, 1989). These operators actually play two roles: for soft-greedification within the Bellman update when bootstrapping off values in the next state (target policy) and to encourage some amount of exploration (in the behavior policy). Soft-greedification within the Bellman update facilitates learning a stochastic soft-optimal policy and can help mitigate the overestimate bias in Q-learning (Song et al., 2019). This changes the target policy being learned. For the second role, regardless of whether a Q-learning update is used or a soft-greedy update, the soft-greedy operator can be used to take actions. Such an approach is actually complementary to directed exploration approaches, like those that learn optimistic values, because it can provide a small amount of additional exploration and so robustness to estimation error in the optimistic values. And for some environments, where only a small amount of exploration is needed, these soft-greedy operators for the behavior actually provide a sufficient level of exploration.

In this work, we re-examine the utility of these operators for exploration, and propose an operator that is more suitable for both roles. In particular, we overcome a few limitations of $\varepsilon$-greedy and softmax. $\varepsilon$-greedy explores in a completely undirected way. Regardless of the agent's estimates of the value of an action, the agent's exploration step is uniformly random. Softmax—a Boltzmann policy on the action-values—is more directed, in that it samples corresponding to the value of the actions. However, softmax is notoriously difficult to tune and suffers from concentrating too quickly

due to the use of an exponential. This concentration overemphasizes actions that appear high-valued under current estimates but are actually suboptimal. This overemphasis can cause softmax to settle on a suboptimal policy, as we reaffirm in our experiments. Further, softmax is not sound when used within the Bellman update: it does not guarantee that the Bellman update is a non-expansion, and so divergence can occur under Sarsa or dynamic programming updates.

There have been some works attempting to improve on these soft-greedy operators. Several works have focused on solving the fact that softmax is not guaranteed to be a non-expansion (Asadi & Littman, 2017; Cesa-Bianchi et al., 2017; Pan et al.). One of these works proposes a soft-greedy operator called mellowmax (Asadi & Littman, 2017) that is guaranteed to be a non-expansion when it is used for soft-greedification. It also effectively tunes the softmax's exploration parameter per state and has been shown to improve stability when it is used for exploration (Kim et al., 2019). However, mellowmax is not a simple heuristic to use, as it requires solving a root finding problem to compute the policy for decision making. To solve this problem they used Brent's method, which is computationally complex (Wilkins & Gu, 2013). An approach called value-difference exploration (Tokic, 2010; Tokic & Palm, 2011) adapts the exploration parameter $\varepsilon$ and the temperature for softmax over time, using the difference in the softmax of the values before and after learning. A later empirical study, however, highlighted that this approach does not perform consistently (Gimelfarb et al., 2020).

Instead of improving these existing operators, we consider a new soft-greedy operator, which we call resmax, based on a probability matching scheme originally developed in the contextual bandit setting (Abe & Long, 1999; Foster & Rakhlin, 2020). This technique is similar to softmax in the sense that it assigns distinct probabilities to actions based on the estimated action-values. However, unlike softmax, the probability for taking each action is determined using its suboptimality gap: the difference between the approximated value of the greedy action and the given action. The policy is inversely proportional to this suboptimality gap, and avoids the use of the exponential that causes softmax to overemphasize actions. We show that it ensures a minimal probability on each action, regardless of the action-values, ensuring all actions are explored. We show that it is a non-expansion, and so combines well with generalized value iteration algorithms. We conclude with an empirical study, across a variety of hard and easy exploration problems, with tabular and deep function approximation. We find that resmax outperforms softmax and $\varepsilon$-greedy, especially when softmax suffers from overestimation.

## 2 BACKGROUND

We model the environment as a discounted Markov Decision Process (MDP) $(\mathcal{S}, \mathcal{A}, \mathcal{R}, P, \gamma)$ where $\mathcal{S}$ is the set of states; $\mathcal{A}$ the set of actions; $\mathcal{R}$ the set of possible rewards; $\gamma \in [0, 1]$ the discount factor; and $P : \mathcal{S} \times R \times \mathcal{S} \times \mathcal{A} \to [0, 1]$ the dynamics function. In a given state $s$, the agent takes action $a$ and transitions to state $s'$ and receives reward $r$ according to probability $p(s', r|s, a)$.

The agent's goal is to learns a policy $\pi : \mathcal{S} \times \mathcal{A} \to [0, 1]$ that maximizes its discounted cumulative reward. To action-value for a state and action, under the policy is

$$q_\pi(s, a) = \mathbb{E}_\pi \left[ \sum_{k=0}^{\infty} \gamma^k R_{t+k+1} | S_t = s, A_t = a \right] \tag{1}$$

The agent attempts to estimate $q_{\pi^*}$, the action value function for the optimal policy $\pi^*$. Many RL algorithms compute these estimates iteratively, using either Q-learning or Expected Sarsa. Both methods use and update to the parameter vector $\theta$ of the form

$$\theta_{t+1} \leftarrow \theta_t + \alpha \delta_t \nabla \hat{q}(s, a, \theta_t)$$

but with different TD errors $\delta_t$. Q-learning uses $\delta_t = r + \gamma \max_{a'} \hat{q}(s', a', \theta_t) - \hat{q}(s, a, \theta_t)$. Expected Sarsa uses $\delta_t = r + \gamma \mathbb{E}_{a' \sim \pi(\cdot|s')} [q(s', a', \theta_t) - \hat{q}(s, a, \theta_t)]$.

To learn these action-values, the agent needs to explore. Two simple exploration methods based on action-value estimates are Boltzmann softmax policy and $\varepsilon$-greedy, defined as

$$\pi_{\text{softmax}}(a \mid s) = \frac{e^{q(s,a)\tau^{-1}}}{\sum\limits_{a' \in \mathcal{A}} e^{q(s,a')\tau^{-1}}} \qquad \pi_{\varepsilon\text{-greedy}}(a \mid s) = \begin{cases} 1 - \varepsilon + \frac{\varepsilon}{|\mathcal{A}(s)|} & \text{if } a = \text{argmax}_a q(s, a) \\ \frac{\varepsilon}{|\mathcal{A}(s)|} & \text{otherwise} \end{cases}$$

with an exploration parameter $\tau > 0$ and $\epsilon \in [0, 1]$. With lower values of $\tau$, softmax is greedier; with higher values of $\tau$, softmax action selection becomes more equiprobable. Resmax and softmax share two essential characteristics: randomness in action selection to guarantee the coverage of the whole state space and a hyperparameter to vary the degree to which the greedy action is chosen.

## 3 THE RESMAX OPERATOR

Softmax and $\varepsilon$-greedy remain two of the most practiced soft-greedy operators for value-based RL algorithms. However, they suffer from several flaws. One of the major problems of $\varepsilon$-greedy is that it ignores the estimates of action-values and assigns a uniform probability to each non-greedy action. This undirected exploration results in wasting time taking actions that the agent might already know are suboptimal. Further, to explore specific parts of the environment, it might need to chain a sequence of random actions, which might be very low probability. For instance, consider the RiverSwim environment (Strehl & Littman, 2008) where an agent receives a minor positive reward if it takes the left action in the initial state, but a much larger reward at the far right of the environment that is harder to reach. To receive this high reward, the agent might need to take several exploratory actions in a row, and so typically gets stuck learning to go left.

Softmax is more directed, in that it assigns the probability of each action corresponding to its action-value. However, if the temperature is not set carefully, softmax will assign an excessively disproportionate probability to the greedy action because this probability is based on exponential q-values. Consequently, softmax often overemphasizes actions that currently have high value estimates, at the expense of exploring other actions. Furthermore, when using the Boltzmann softmax operator in the Expected Sarsa update, for the expectation of the value in the next state, this operation is not guaranteed to be a non-expansion (Littman, 1996; Littman & Szepesvári, 1996). As mentioned earlier, mellowmax operator was designed to fix this non-expansion issue (Asadi & Littman, 2017), but mellowmax policy is not as straightforward to use as softmax policy because an optimization problem needs to be solved to select the temperature in each state.

Our goal is to gain the best of both worlds: using the information provided by action values without suffering from the two issues with softmax. For this purpose, we propose resmax. For greedy action $b = \arg\max_a q(s, a)$ and non-greedy action $a$, the resmax policy probability is

$$\pi(a \mid s) = \frac{1}{|\mathcal{A}| + \eta^{-1}(q(s, b) - q(s, a))} \qquad \pi(b \mid s) = 1 - \sum_{a \neq b} \pi(a \mid s) \qquad (2)$$

where the exploration parameter $\eta > 0$ can be thought of as the *exploration pressure*. Large $\eta$ values push towards exploration whereas low values result in more exploitation— are greedier. When $\eta \to \infty$ the policy is uniformly random and when $\eta \to 0$ the policy is greedy with respect to $q(s, b)$.

This operator is efficient to compute and easy to use for exploration or soft-greedification. To use it with Q-learning or DQN, the agent simply samples actions from the resmax policy in Equation 2. To use it for greedification, we can simply use the corresponding resmax operator

$$\mathrm{rm}(q(s, \cdot), \eta) \doteq \sum_{a' \in \mathcal{A}} \pi(a' \mid s) q(s, a') \qquad (3)$$

in Sarsa or Expected Sarsa. Note that as $\eta$ goes to infinity, this operator becomes the max operator (please refer to Section A.2 in the appendix for details).

## 4 RESMAX AVOIDS OVEREMPHASIS AND ENCOURAGES EXPLORATION

In this section, we first confirm that resmax is guaranteed to provide sufficient exploration to get satisfactory convergence properties for certain families of RL algorithms. The resmax policy can either be used off-policy, on top of action-values learned with Q-learning, or on-policy. In either case, it is key to ensure we have sufficient exploration. Note that this first role is separate from the second role of the resmax operator, which is to ensure we have a non-expansion within the Bellman update; we discuss this property in Section 5. Then, we show that resmax avoids overemphasis by conducting experiments in a small environment.

Exploration strategies should satisfy certain fundamental properties to make sure that algorithms such as Q-learning and Sarsa converge to the optimal value (Singh et al., 2000; Watkins & Dayan, 1992). A key property is that each state-action pair should be visited infinitely many times during continual learning. To show that resmax satisfies this property, we prove that the probability of taking all of the actions will be higher than zero during learning for any bounded action-values (please refer to Section A.1 in the appendix for the proof).

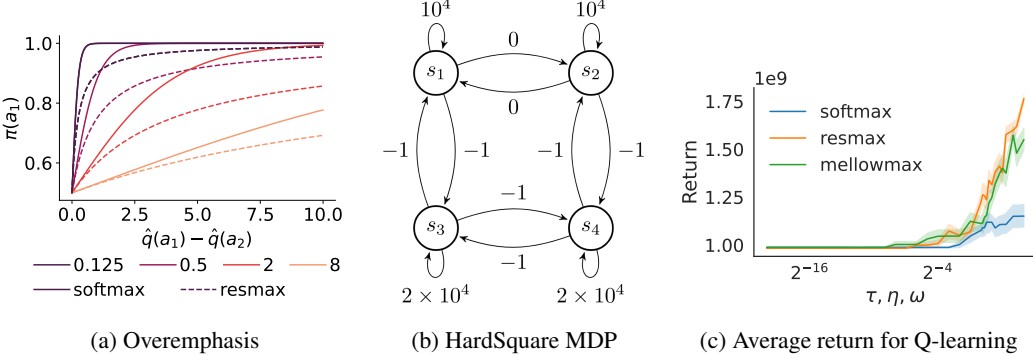

(a) Overemphasis       (b) HardSquare MDP       (c) Average return for Q-learning

Figure 1: (a) Visualization of overemphasis in softmax and resmax over a range of parameters. As the suboptimality gap increases, softmax is quicker than resmax to place high probability on action $a_1$. Values of $\tau$ and $\eta$ are given in the legend. Dotted lines are resmax and solid lines are softmax. (b) The HardSquare MDP: a simple MDP with four states and three actions and $\gamma = 0.95$. The edges show a deterministic transitions and rewards given an action. The agent starts in states $s_1$ and $s_2$ with equal probability. (c) Performance of resmax, mellowmax and softmax on HardSquare. The x-axis is the hyper-parameter choices for each operator. The y-axis is the average return over the total time steps. Softmax will get stuck in states $s_1$ and $s_2$, whereas resmax can escape from the initial states and successfully explore towards higher value states. Results are shown for $\tau, \eta \in \{2^{-20}, 2^{-3}, ..., 2^4\} \cup \{1/0.1, 1/0.2, ..., 1/0.9\}$.

A serious issue with using the softmax for exploration comes from the fact that it uses exponents within its formulation, which can assign an overly disproportionate probability to current estimate of the optimal action. We call this phenomenon *overemphasis*, which can cause value-based exploration using softmax to get stuck in local minima. This effect is analogous to the *softmax gravity well* described by Mei et al. (2020) in the context of policy gradient methods.

To illustrate, consider a bandit problem with two actions, $a_1$ and $a_2$. Suppose that $a_2$ is the optimal action, but due to stochasticity in the learning process, the estimate $\hat{q}(a_1) > \hat{q}(a_2)$. We visualize the resultant softmax policy $\pi(a_1)$ over a range of $\tau$ for softmax and $\eta$ for resmax in Figure 1a. Here we set $\hat{w}(a_1) \in [-5, 5]$, and $\hat{w}(a_2) = -5$. In most choices of the temperature $\tau$, if $\hat{w}(a_1)$ is sufficiently larger than $\hat{w}(a_2)$, the outputs of softmax $\pi(a_1)$ will be near 1. Thus, it may take a very large number of steps to finally choose action $a_2$ and explore another action. If instead the policy is chosen according to resmax, the probability assigned to the greedy action is proportionate to the suboptimality gap and therefore less extreme. Consequently exploration of $a_2$ becomes more likely.

The severity of the softmax overemphasis problem is further demonstrated in the HardSquare MDP (Figure 1b). The agent starts in the state $s_1$ and $s_2$ with an equal probability and all actions are deterministic. The agent can stay in the start states for a reward of $10^4$, or move to $s_3$ and $s_4$, which gives a short term reward of -1, but in these states the agent can receive a larger reward of $2 \times 10^4$. We use tabular Q-learning with either resmax or softmax as the exploration heuristic on this problem. We initialize $\hat{Q}(s, a) \leftarrow 0$, $\forall (s, a) \in \mathcal{S} \times \mathcal{A}$. Results are averaged over 30 runs. As shown in Figure 1c, the softmax policy gets stuck in the initial states $s_1$ and $s_2$ due to its tendency to overemphasize the approximated optimal action. On the contrary, resmax can escape from the initial states and successfully explore towards the optimal solution. We include mellowmax in our results for hardsquare and note that it performs similarly to resmax. However, due to the optimization procedure inherent in mellowmax, its runtime is on average $23\times$ longer than resmax. This prohibitively expensive runtime prevented us from further considering mellowmax in our results.

## 5 RESMAX IS A NON-EXPANSION

The second key property of resmax is that it is a non-expansion. The non-expansion property ensures that Generalized Value Iteration (GVI) algorithms, such as value iteration, will converge to a *unique* fixed point (Littman & Szepesvári, 1996; Asadi & Littman, 2017). Without the non-expansion property—as is the case for the softmax operator—an operator may converge to multiple fixed points, even in the tabular setting, which in turn leads to misbehaviors in both learning and planning.

Figure 2a shows a simple two state MDP (taken from Asadi & Littman (2017)) where the action values of softmax may not converge, as shown in 2b. Resmax, since it is a non-expansion, converges for all settings of $\eta$. We show in Figure 2c that resmax indeed converges under a setting of $\eta$ that provides similar exploration to softmax.

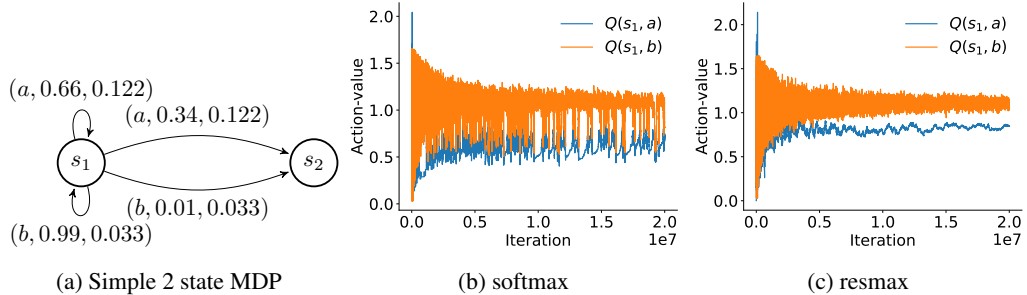

(a) Simple 2 state MDP     (b) softmax       (c) resmax

Figure 2: The non-expansion property is important for converges under GVI. On the MDP shown in (a) the action-values, if using softmax in the expected sarsa update (with $\tau = 1/16.55$), do not converge (b). We can see that resmax does converge in (c) with $\eta = 0.000085$. We use a step size $\alpha$ at time $t$ to be $1/(\lfloor t/100,000 \rfloor + 1)$, which meets the usual conditions for stochastic approximation to converge (Sutton & Barto, 2018). Note that the tuples in (a) are of the form (action, probability of transition given action, reward). Results are smoothed with a window size of 10, as in Asadi & Littman (2017).

We now show that the resmax operator is indeed a non-expansion.

**Property 5.1.** *The resmax operator is a non-expansion, i.e., for any two vectors $\vec{x}, \vec{y} \in \mathbb{R}^d$ and exploration pressure $\eta \geq 0$ we have,*

$$|\text{rm}(\vec{x}, \eta) - \text{rm}(\vec{y}, \eta)| \leq \max_{i \in [d]} |x_i - y_i| \tag{4}$$

*Proof.* The resmax operator is defined as follows

$$\text{rm}(\vec{x}, \eta) = \sum_{i \in [d], i \neq j} \frac{x_i}{d + \frac{1}{\eta}(x_j - x_i)} + x_j \left[ 1 - \sum_{i \in [d], i \neq j} \frac{1}{d + \frac{1}{\eta}(x_j - x_i)} \right]$$

$$= \sum_{i \in [d], i \neq j} \frac{x_i - x_j}{d + \frac{1}{\eta}(x_j - x_i)} + x_j \tag{5}$$

where $j \doteq \arg \max_i x_i$. To show that resmax is a non-expansion, we need a special instance of Theorem 1 of Paulavičius & Žilinskas (2006) which states the following. For Lipschitz function $f(x)$, $f : \mathbb{R}^d \to \mathbb{R}$,

$$|f(x) - f(y)| \leq L_1 \|x - y\|_\infty \tag{6}$$

where $L_1 = \sup\{\|\nabla f(x)\|_p : x \in D\}$ is the Lipschitz constant, $D$ is a compact set, and $\nabla f(x) = (\partial f/\partial x_1, ..., \partial f/\partial x_d)$ is the gradient of the function $f(x)$. Therefore, we need to show that the $\ell_1$ norm of the gradient of resmax is less than or equal to 1, namely $L_1 \leq 1$. To compute the gradient of $\text{rm}(\vec{x}, \eta)$ for $i \neq j$, let $\delta_i = x_j - x_i$ to get

$$\frac{\partial}{\partial x_i} \text{rm}(\vec{x}, \eta) = \frac{d\eta^2}{(d\eta + \delta_i)^2}$$

We now compute $\partial \text{rm}(\vec{x}, \eta)/\partial x_j$,

$$\frac{\partial}{\partial x_j}\text{rm}(\vec{x}, \eta) = 1 - \sum_{i \in [d], i \neq j} \frac{d\eta^2}{(d\eta + \delta_i)^2}$$

Notice now that

$$\frac{d\eta^2}{(d\eta + \delta_i)^2} = \frac{d\eta^2}{d^2\eta^2 + 2d\eta\delta_i + \delta_i^2} \leq \frac{d\eta^2}{d^2\eta^2} = \frac{1}{d}.$$

where the inequality holds because all the terms in the denominator are positive. Therefore, we know that the partial derivative for $j$ is positive. Further, the partial derivatives for $i \neq j$ are also positive, so taking the $\ell_1$ norm corresponds to simply summing up those terms

$$\sum_{i \in [d], i \neq j} \frac{d\eta^2}{(d\eta + \delta_i)^2} + 1 - \sum_{i \in [d], i \neq j} \frac{d\eta^2}{(d\eta + \delta_i)^2} = 1$$

Since this is true for any $\vec{x}$, we know that $L_1 = 1$, and so resmax is a nonexpansion. □

This result also lets us show that the sequence of policies generated by approximate policy iteration under the resmax operator converges to a unique limiting policy regardless of the choice of the initial policy $\pi_0$. This follows from Theorem 1 of Perkins & Precup (2002), which simply requires that the operator be Lipschitz continuous with an $\epsilon$-soft policy. Policy $\pi$ is $\epsilon$-soft if $\pi(a|s) > \epsilon$. As shown in Section A.1, the resmax policy is $\epsilon$-soft. By the non-expansion property of resmax, we can also say the resmax operator is Lipschitz continuous (Asadi & Littman, 2017).

## 6 EXPERIMENTS

In this section, we empirically investigate resmax through a series of experiments. We first use a tabular hard exploration environment, RiverSwim, to show that resmax promotes more exploration than softmax since it does not overemphasize. Next, we show that resmax scales to use with function approximation by benchmarking performance on standard environments. We additionally study the performance of resmax and softmax when varying the replay buffer size of DQN. Details on implementation and computational infrastructure are given in the appendix.

### 6.1 RESMAX PROMOTES MORE EXPLORATION THAN SOFTMAX

In Section 4 we showed softmax can overemphasize actions that appear high-valued according to its approximate estimates, and so converge to a suboptimal solution. Resmax does not have the same overemphasis issues, and promotes more exploration. To illustrate, we use the RiverSwim environment (Strehl & Littman, 2008). RiverSwim consists of 6 states arranged in line, of which the agent starts in the leftmost. This state has a small reward but the rightmost state may give a comparatively large reward. In order to reach this state the agent must traverse the MDP from left to right, fighting against a "current" which causes moving right to oftentimes fail. Reaching this rightmost state requires smart exploration. We use RiverSwim with a fixed horizon, meaning that after 20 actions the agent will be returned to the start state, in effect making the environment more difficult. All algorithms are run for 800,000 time steps.

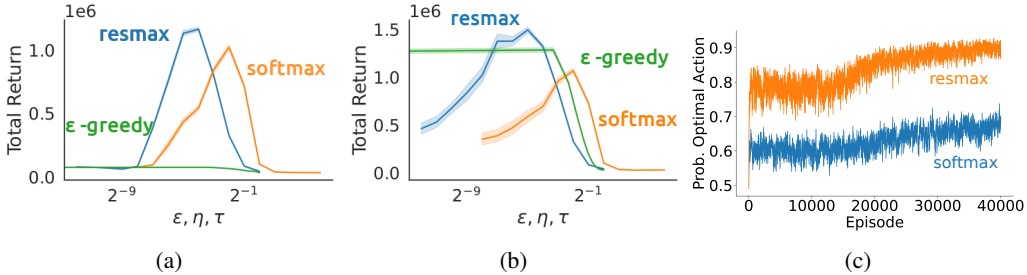

Figure 3: (a) Hyperparameter sensitivity for RiverSwim. (b) Hyperparameter sensitivity for Stochastic-reward RiverSwim. For both (a) and (b) means and standard errors are shown for 30 runs for each parameter setting. The x-axis is plotted with a log scale, and shows the value of exploration parameters. Note that the horizontal sensitivity curve for (c) The probability of selecting the optimal action (right) from the initial state during the first step in the episode in Stochastic-reward RiverSwim. Results are averaged over 100 runs and are for optimal choices of hyperparameters in Stochastic-reward RiverSwim: $\eta = 2^{-5}$, $\tau = 2^{-2}$ and $\varepsilon = 0.1$.

**Experiments with Misleading Rewards**

Misleading rewards are rewards that, when pursued, lead the agent away from potentially higher reward. Following such a signal can be viewed as over-eager exploitation; ignoring the misleading signal and exploring further can lead to much higher long-term reward. HardSquare (Figure 1b) is one example of an environment with misleading rewards. RiverSwim is also such an example: the reward on the leftmost state distracts the agent from the much higher reward on the right.

We ran softmax, resmax and $\varepsilon$-greedy coupled with Expected Sarsa on RiverSwim across a broad range of hyperparameters ($\eta$, $\tau$ and $\varepsilon$). The hyperparameter ranges were chosen to match the technique, for example softmax requires relatively larger values of $\tau$ due its exponents. Results are shown in Figure 3a, where we measure algorithm performance by the total online return across these hyperparameters. We find that resmax and softmax perform similarly, however resmax achieves a higher total return under optimal hyperparameter settings than softmax, indicating it is more capable of escaping the pull of the left state. $\varepsilon$-greedy fails to explore at all, and its total return is hardly visible for any values of $\varepsilon$.

**Interaction with Stochasticity in Rewards**

Stochasticity in rewards can help alleviate some of the misleading reward problem. Uncertainty in the rewards can prevent over-exploitation of sub-optimal actions and help drive exploration. We replaced the deterministic positive rewards in RiverSwim with rewards drawn from a Gaussian distribution with the mean being the original reward and variance of 1. Results are shown in 3b. All three techniques see a reduction in hyperparameter sensitivity, but resmax and $\varepsilon$-greedy increase their total return under optimal parameter settings, whereas softmax does not. With stochastic rewards, softmax still assigns too great a weight on sub-optimal actions, as shown in Figure 3c. Softmax learns to take the optimal action in the initial state at a slower rate than resmax, indicating that softmax is concentrating too heavily on misleading rewards.

## 6.2 RESMAX WITH FUNCTION APPROXIMATION

Many RL algorithms with function approximation use $\varepsilon$-greedy as a default method to add exploration. Resmax, in order to be considered a general-purpose method, should also scale to this setting. To show this, we conduct experiments in deep RL setting. We first study the stability of directed exploration techniques, resmax, and softmax, across different replay buffer sizes on two standard OpenAI gym environments (Brockman et al., 2016), namely Acrobot-v1 and LunarLander-v2, and Sparse MountainCar (Kumaraswamy et al., 2018). Then, we extend our deep RL experiments to eight large-scale Atari environments.

### 6.2.1 STABILITY OF RESMAX ACROSS DIFFERENT REPLAY BUFFER SIZES

Here, we study the effect of varying the replay buffer size on the performance of the resmax and softmax operator to compare the performance of these two soft-greedy operators that do more directed exploration. For implementing the Sparse MountainCar environment that is commonly used for benchmarking exploration algorithms, we modified the reward system of OpenAI implementation of MountainCar environment by setting the reward in all of the states to $0$, except in the states at the top of the hill. We have also increased the maximum number of steps in Sparse MountainCar to $5,000$. All other environments have been used exactly as they are implemented in OpenAI gym. We use Deep Expected Sarsa for running our experiments in this section, which is the same as DQN (Mnih et al., 2015) except that instead of the maximum over next state–action pairs to compute targets it uses the expected value. We selected this algorithm because it uses resmax operator presented in Equation 3 to compute the targets. So, in this way, we can also see how operator performs in the deep RL setting compared to softmax. Exact hyperparameter settings are given in the appendix.

The results for these experiments are presented in Figure 4. Resmax achieves a better performance than softmax on LunarLander and Sparse MountainCar environments that require a higher level of exploration and performs similar to softmax on Acrobot environment. In terms of stability, whereas softmax is highly unstable in LunarLander environment and performs poorly with the low replay buffer size, resmax is able to maintain a good performance with $\eta = 2^{-8}$ across different sizes of the replay buffer. Both operators gain the same stability on Sparse MountainCar and Acrobot environments. We also ran these experiments on a larger set of hyperparameters for each operator and included epsilon-greedy. The plots for these larger set of experiments are presented it in the Appendix.

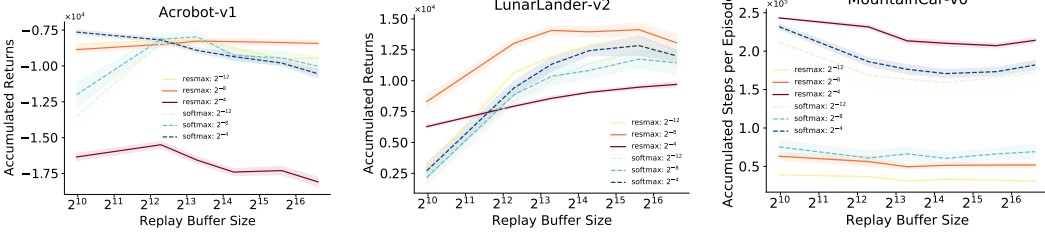

Figure 4: Replay buffer size sensitivity plots describing the effect of different sizes of replay buffer on the performance of the agent for Deep Expected Sarsa. Results shown are averaged over 30 runs for each parameter setting. The step-size value is set to $10^{-4}$ for all of the environments. Lower values are better in Sparse MountainCar plot and higher values are better in the other two plots. The value in the legends indicate the value of the exploration parameter of each soft-greedy operator.

### 6.2.2 EXPERIMENTS ON ATARI ENVIRONMENTS

We use the DQN algorithm (Mnih et al., 2015) for our Atari experiments. We investigate the sensitivity of resmax, softmax, and $\varepsilon$-greedy to their exploration parameters and compare their best-performing instances to each other. The results of these experiments are shown in Figure 5. As evident from these graphs, $\varepsilon$-greedy fails to reach a similar or better performance than both resmax and softmax across all three environments. Resmax performs similar to softmax on the Asterix environment. However, it outperforms softmax in Freeway by a large margin and gains a higher return in the Breakout. These graphs also show that resmax is much more stable across a large range of its exploration parameters while this is not the case for softmax and $\varepsilon$-greedy. Additional results for five other atari environments, three of which are considered hard-exploration (Taiga et al., 2020), are presented in section E.4 in the appendix. These results provide further evidence that resmax achieves highly better performance over the baselines.

It is good to mention that we chose Freeway because it is a hard-exploration environment (Taiga et al., 2020). As our tabular experiments in hard-exploration problems like RiverSwim have shown, resmax outperforms other baselines in this set of environments. Freeway is similar to RiverSwim in the sense that chicken should persistently go up to reach the top of the screen to get a positive reward while

taking into account the stochasticity in the transition function that is caused by the cars that can crash into the chicken at any moment.

Figure 5: Bar charts describing the effect of different level of exploration on the performance of soft-greedy operators in DQN setting and learning-curves presenting the best performance across different exploration parameters. Results shown are averaged over 10 runs for each parameter setting. The step-size value is set to $10^{-4}$ across all the environments. The hyperparameters corresponding low, medium and high exploration used in the x-axis are given in the appendix.

## 7 Conclusion

Soft-greedy operators, including $\varepsilon$-greedy, softmax and resmax, continue to play an important role in reinforcement learning. They serve dual purposes: they function as soft-greedification within the Bellman update and induce a basic level of exploration; and hence, are valuable for both on-policy and off-policy learning. While simple, these operators are complementary to function approximation and more focused exploration techniques. We propose a new soft-greedy operator, called resmax. Its properties are both theoretically desirable and perform well empirically. Unlike softmax, resmax is a non-expansion, regardless of the hyperparameter settings and thus converges under generalized approximate policy iteration. Moreover, resmax ensures state-action space coverage and it avoids softmax's fundamental issue of overemphasis. Our empirical results show that resmax encourages more exploration than softmax in RiverSwim, since it does not overemphasize and also explores more efficiently than $\varepsilon$-greedy.

This paper proposes resmax in its simplest form, so there are many avenues for future research. As resmax is a new operator to reinforcement learning, it deserves further benchmarking and experimentation in both simple and complex environments in order to shed more light on when this operator is useful. Another natural direction for future work is to explore adaptation and normalization techniques which are suitable for resmax. Just like decay schedules for $\varepsilon$, schedules or adaption schemes for resmax could allow more exploration in early layer, and allow greedier policies later.

## 8 Reproducibility Statement

All the experiments in this paper are reproducible and can be reproduced by the code made available in the submitted supplementary materials. The instructions for reproducing these experiments are given in the Readme files in the provided source code. Also, all the major proofs of this paper along with their assumptions are presented in the main text of the paper. Finally, no datasets are used for this paper and all the environments that are used are freely available.

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

# A ADDITIONAL PROPERTIES OF RESMAX

## A.1 STATE-ACTION SPACE COVERAGE

Exploration strategies should satisfy certain fundamental properties to make sure that algorithms such as Q-learning and Sarsa converge to the optimal value (Singh et al., 2000; Watkins & Dayan, 1992). A key property is that each state-action pair should be visited infinitely many times during continual learning. To show that resmax satisfies this property, we prove that the probability of taking all of the actions will be higher than zero during learning for any bounded action-values.

**Property A.1.** *Assume that there exists $q_{max}$ such that $\forall s, a, |q(s, a)| \le q_{max}$. The probability of taking any non-greedy action $a$ and a greedy action $b$ is bounded as follows:*

$$0 < \frac{1}{|\mathcal{A}| + 2\eta^{-1}q_{max}} \le \pi(a \mid s) \le \frac{1}{|\mathcal{A}|} \le \pi(b \mid s) < 1 \tag{1}$$

*Proof.* First we determine the upper-bound and lower-bound for non-greedy actions $a$ and then proceed to show this for the greedy action $b$. By analyzing Equation 2, it is clear that its lowest-value will be obtained only when the difference between $q(s, b)$ and $q(s, a)$ is at its highest. We therefore get the following lower bound on $\pi(a \mid s)$

$$\pi(a \mid s) = \frac{1}{|\mathcal{A}| + \eta^{-1}(q(s, b) - q(s, a))} \ge \frac{1}{|\mathcal{A}| + \eta^{-1}(q_{max} - (-q_{max}))} = \frac{1}{|\mathcal{A}| + 2\eta^{-1}q_{max}} > 0$$

Furthermore, the highest possible value for $\pi(a \mid s)$ for a non-greedy action $a$ is $1/|\mathcal{A}|$, which occurs when $q(s, a) = q(s, b)$. Based on Equation 2, when the probability of non-greedy actions reaches its highest value, the probability of greedy action $b$ will be at its lowest, which is similarly $1/|\mathcal{A}|$. Therefore, we have $\pi(s \mid a) \le 1/|\mathcal{A}| \le \pi(b \mid s) < 1$, completing the proof. ☐

## A.2 RESMAX DOES MAXIMIZATION

It can be easily shown that the Expected Sarsa update of resmax, represented in Equation 3, can do maximization (i.e., give the highest probability to the greedy action) when $\eta$ goes to 0. We first show how $\pi(a \mid s)$ for non-greedy actions will change when this happens:

$$\lim_{\eta \to 0} \pi(a \mid s) = \lim_{\eta \to 0} \frac{1}{|\mathcal{A}| + \eta^{-1}(q(s, b) - q(s, a))} = 0$$

This equality will hold as long as $q(s, b) \ne q(s, a)$.

Considering this, we can derive Equation 3 when $\eta$ goes to infinity as follow:

$$\lim_{\eta \to 0} \text{rm}(q(s, \cdot), \eta) = \lim_{\eta \to 0} \sum_{a \ne b} \pi(a \mid s)q(s, a)$$
$$+ (1 - \sum_{a \ne b} \pi(a \mid s))q(s, b)$$
$$= 0 - (1 - 0)q(s, b) = q(s, b)$$

Since $q(s, b)$ is the action-value of the greedy action, resmax can do maximization. It is also interesting to note that when $\eta$ goes to $\infty$, the generated policy will be equiprobable thus Expected Sarsa update of resmax will average all the action-values. So, this operator can make a balance between q-learning update and update with equiprobable policy by tuning the value of $\eta$ like softmax and mellowmax.

# B COMPUTATIONAL INFRASTRUCTURE

We ran our experiments on a compute cluster. Each job used a single CPU core, except for atari experiments that we used GPUs. The compute cluster allocated CPUs based on availability. The possible options were 2.1Ghz Intel CPUs with model numbers E5-2683 V4 Broadwell, E7-4809 V4 Broadwell, or Platinum 8160F Skylake, as well 2.4Ghz Intel Platinum 8260 Cascade Lake. For

GPU experiments, we used V100 Volta GPU. We also requested 400MB for the tabular setting. In the case of deep RL, we requested 4GB for low-scale environments and 16GB of memory for Atari environments. Different algorithms and exploration heuristics within one environment used the same configurations of resources.

## C    LOGGING PROCEDURE

To save returns and steps per episode, we average returns or the number of steps per episode for all of the episodes that have been finished in a specific number of steps that we call log-interval. To elaborate, returns and the number of steps per episode for all the episodes that are finished in the log interval will be accumulated and averaged. We only store this final averaged value. For instance, if we set the total number of steps to $100,000$ and define a log-interval of $1,000$, $100$ values will be stored. This way of storing the results of our experiments can save us both memory and space. At the same time, the stored results are proportional to the performance of each of the employed algorithms. We use log-interval of $1,000$ for all our tabular experiments and log-interval of $10,000$ for all the DQN experiments, except in Atari experiments that we use log-interval of $100,000$.

## D    HYPERPARAMETER STUDY

In this section, we present the hyperparameters that are used in our experiments and the reason for selecting them. One of the hyperparameters we needed to set fairly across different soft-greedy operators were their respective exploration parameters, such as $\eta$ and $\tau$. To do this, we swept over a large set of hyperparameters for each soft-greedy operator to make sure that the exploration parameter with the near-best performance resides in this set.

Our tabular experiments used $\eta \in \{2^{-12}, 2^{-11}, ..., 2^0\}$, $\tau \in \{2^{-8}, 2^{-7}, ..., 2^4\}$ and $\varepsilon \in \{0, 0.1, ..., 1\}$. Softmax and resmax required different ranges of hyperparamters since softmax uses exponents, and would therefore need larger values of $\tau$ than resmax does for $\eta$.

We plotted the hyperparameter sensitivity in the function apprimation settings across a large set of hyperparameter in the range of $2^{-24}$ to $2^8$ for different environments. Our experiments showed that all the best exploration parameters for these operators are located somewhere between $2^{-24}$ and $1$, except for softmax in CartPole-v0 environment that works best with values lower than $1$. For $\varepsilon$-greedy on the other hand, we noticed that $\varepsilon = 0.1$ always results in a near-best performance, so we only tried several values around it to see how increasing exploration changes its performance. Based on this, we have chosen the exploration parameters for each operator and presented its hyperparameter sensitivity plots in the appendix.

In the deep RL setting, we use the DQN algorithm. We chose a fixed set of parameters that work well across all three benchmark environments. These parameters are presented in Appendix Table 1. We swept over three different step sizes across all our experiments: $0.0005$, $0.0001$, $0.00005$. Our experiments with these step sizes show that step size of $0.0001$ works best across all the low-scale and large-scale environments. We present the results in this paper based on these step sizes. Furthermore, we ran our first set of DQN experiments with $\eta, \tau \in \{2^0, 2^2, ..., 2^{16}\}$ and $\varepsilon \in \{0.1, 0.3, 0.5, 0.7\}$. Then, based on these results, we have cho

When we were choosing low, medium, and high exploration values, we selected a lower set of values for softmax compared to resmax because softmax exponentially reduces the degree of exploration when we increase $\tau$ but this is not the case for resmax. In our DQN experiments, we used $2^0$, $2^8$, $2^{16}$ for softmax and $2^8$, $2^{16}$, $2^{24}$ for resmax and mellowmax as high, medium, and low exploration values. For $\epsilon$-greedy, we used $0.1$, $0.3$, and $0.5$. It is good to mention that these set of values were chosen based on several experiments that we ran on Atari environments showing that these operators work best with these set of values when Atari environments need low, medium, and high level of exploration.

| Parameter Name | Fixed Value |
|---|---|
| Optimizer | Adam |
| $\beta_1$ | 0.9 |
| $\beta_2$ | 0.999 |
| $\epsilon$ for Adam | $10^{-8}$ |
| Batch size | 64 |
| Buffer size | $50,000$ |
| Number of training steps per iteration | 1 |
| Target network update frequency | $1,000$ |
| Number of steps before learning starts | $5,000$ |
| Number of hidden layers | 2 |
| Number of Neurons in each layer | 64 |
| $\gamma$ | 0.99 |

Table 1: The fixed parameters used to run DQN experiments

## E    ADDITIONAL RESULTS

In this section, we present additional results for our experiments in the tabular and Deep RL settings.

### E.1    TABULAR

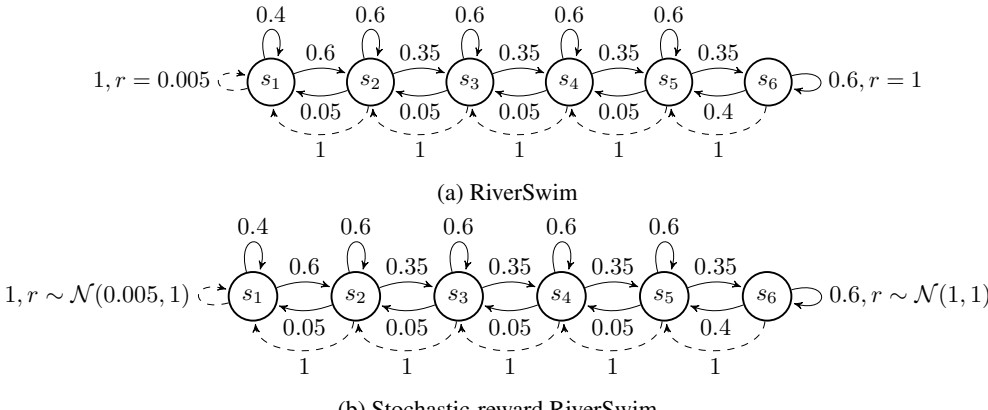

(a) RiverSwim

(b) Stochastic-reward RiverSwim

Figure 1: Diagram of RiverSwim and Stochastic-reward RiverSwim. Dotted lines and solid lines show the transitions and probabilities for the left and right actions, respectively. Diagram adapted from Osband et al. (2013).

## E.2   DEEP RL

In this section, we first present learning curves of best hyperparameters for each soft-greedy operator along with hyperparameter sensitivity plots of each operator and its sensivitiy plots for different replay buffer sizes. Then, we discuss our results on five atari environments.

## E.3   REPLAY BUFFER STUDY

Here first we go through the performance of these operators based on different exploration parameters. Then, we present the plots for their stability across different replay buffer sizes.

The results for our first set of experiments are presented in Appendix Figures 2 and 3. Overall, softmax and resmax are similar in terms of hyperparameter sensitivity and best performance, except in the Sparse MountainCar environment that resmax is much less sensitive compared to softmax and its best performing instance outperforms softmax. $\varepsilon$-greedy, on the other hand, achieves a comparative performance with $\varepsilon = 0.1$ in comparison to the best instance of resmax. In the case of LunarLander-v2, whereas the best instance of $\varepsilon$-greedy surpasses resmax in the beginning of learning, the final performance of resmax is slightly better than $\varepsilon$-greedy.

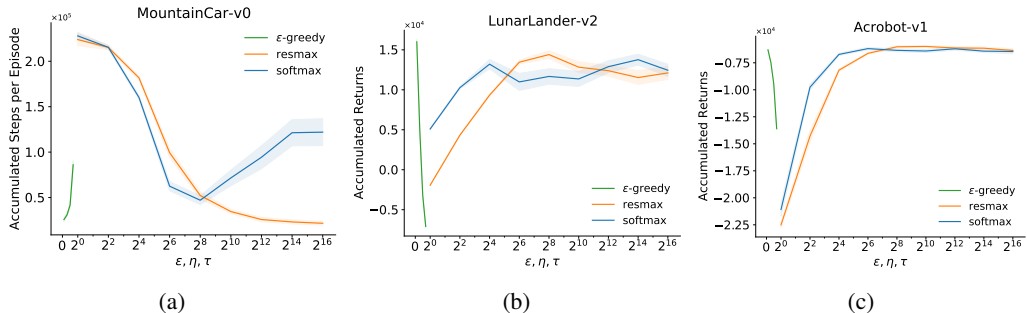

Figure 2: Hyperparameter sensitivity for DQN. Results shown are averaged over $30$ runs for each parameter setting. The x-axis is plotted with a symmetrical log scale.

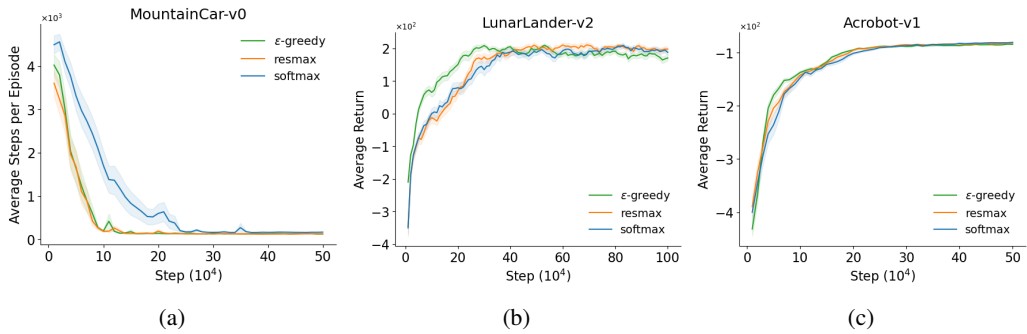

Figure 3: Learning curves for best performing hyperparameter settings for DQN. Best settings (based on average return) are shown in the top left of each subplot. Results are averaged over $30$ runs. $\varepsilon$-greedy consistently reaches its best performance with $\varepsilon = 0.1$ across all three environments. Resmax achieves its best performance with $\eta$ values of $2^{16}$, $2^8$, and $2^{10}$ in Sparse MountainCar, LunarLander-v2, and Acrobot-v1 respectively. Softmax performs best with $\tau$ values of $2^8$, $2^{14}$, and $2^6$ in Sparse MountainCar, LunarLander-v2, and Acrobot-v1 respectively.

The results describing the effect of different sizes of replay buffer on the performance of the agent for Deep Expected Sarsa algorithm is presented in Figure 4.

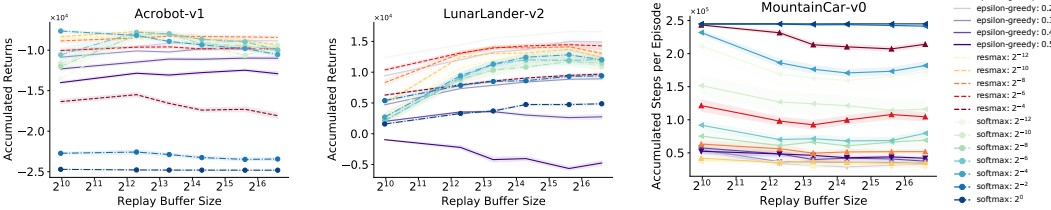

Figure 4: Replay buffer size sensitivity plots describing the effect of different sizes of replay buffer on the performance of the agent for DQN. Results shown are averaged over 30 runs for each parameter setting. The step-size value is set to $10^{-4}$ for all of the environments. Lower values are better in Sparse MountainCar plot and higher values are better in the other two plots.

### E.4 ADDITIONAL ATARI RESULTS

Here we present the results of our experiment on five atari environments, including Pong, SpaceInvaders, Gravitar, Pitfall, and Venture. First, we go through our results on three of these environments that are considered hard-exploration: Gravitar, Pitfall, and Venture. Then, we discuss our results on Pong and SpaceInvaders environments.

The results for hard-exploration atari environments are presented in Figure 5. As evident from this graph, resmax surpasses the performance of softmax and $\varepsilon$-greedy policies on Gravitar and Venture. Especially in Venture, there is a huge performance gap between resmax and other baselines. However, the best performing instance of all the soft-greedy policies reaches similar performance with $\varepsilon$-greedy performing slightly better than softmax. These results show that resmax is a promising technique to be used even for hard-exploration problems.

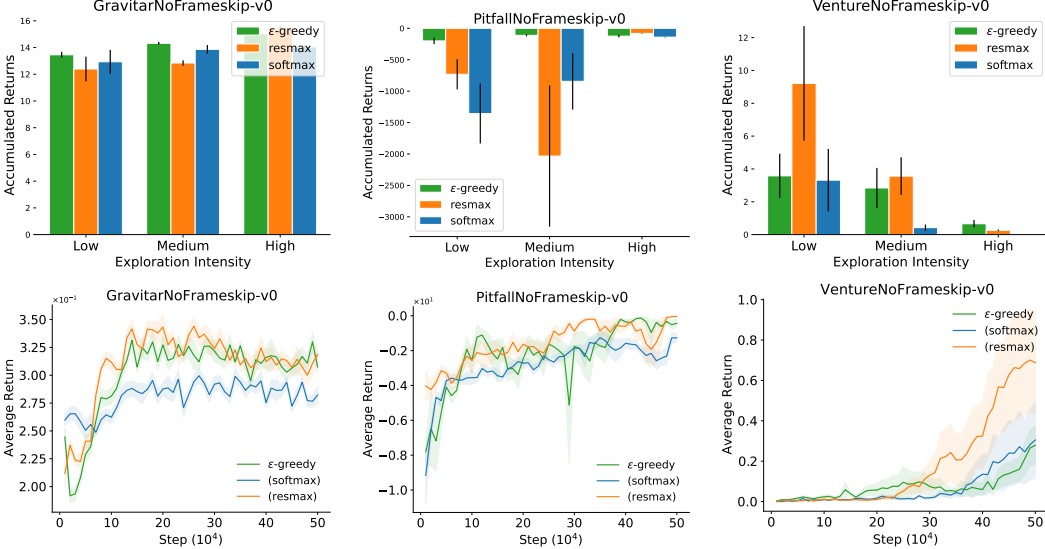

Figure 5: Bar charts for hard exploration atari environments describing the effect of different level of exploration on the performance of soft-greedy operators in DQN setting and learning-curves presenting the best performance across different exploration parameters. Results shown are averaged over 10 runs for each parameter setting. The step-size value is set to $10^{-4}$ across all the environments. The hyperparameters corresponding low, medium and high exploration used in the x-axis are given in the appendix.

Finally, the results on Pong and SpaceInvaders environments are shown in Figure 6. These results follow a similar trend to the results that we have for hard exploration environments, with resmax being highly competitive to the provided baselines.

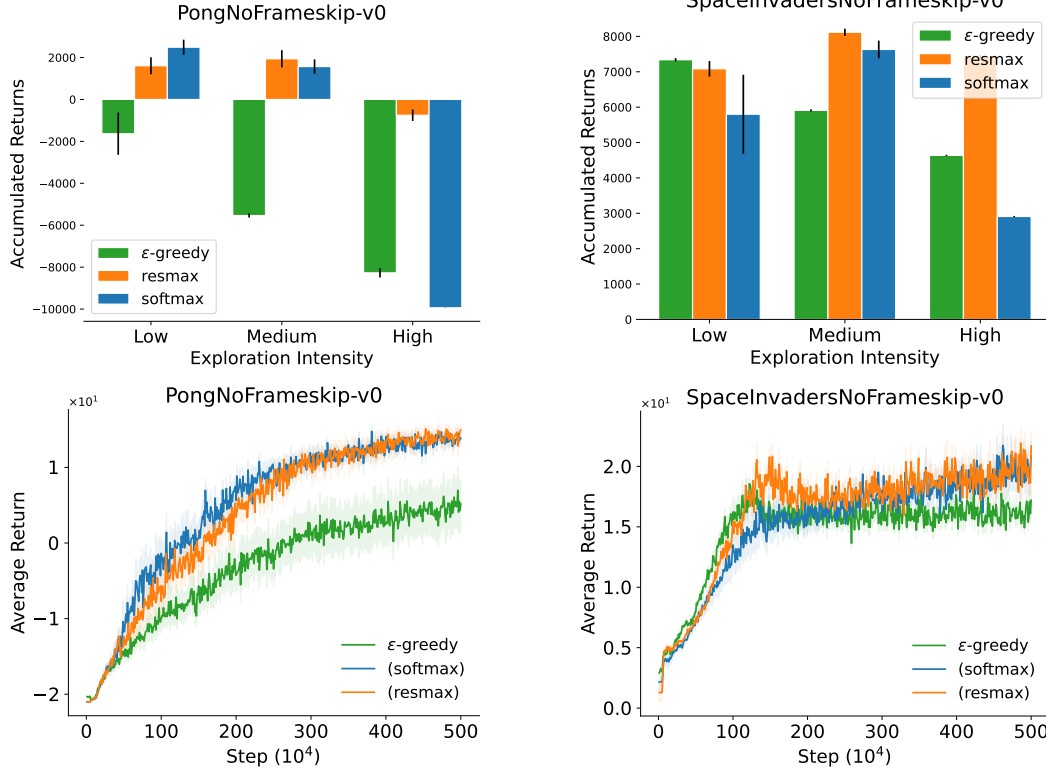

Figure 6: Bar charts describing the effect of different level of exploration on the performance of soft-greedy operators in DQN setting and learning-curves presenting the best performance across different exploration parameters. Results shown are averaged over 10 runs for each parameter setting. The step-size value is set to $10^{-4}$ across all the environments. The hyperparameters corresponding low, medium and high exploration used in the x-axis are given in the appendix.

