# OpenReview forum: "Resmax: An Alternative Soft-Greedy Operator for Reinforcement Learning"
_ICLR.cc/2022/Conference — ICLR 2022 Submitted_

### Official Review · Reviewer_ypnG · 2021-10-26

**Correctness:** 3
**Technical Novelty And Significance:** 3
**Empirical Novelty And Significance:** 3
**Recommendation:** 6
**Confidence:** 4

**Main Review:**

Strengths:
- The article is clear and easy to follow.
- The proposed policy is simple and fast to compute. Much simpler and faster than mellowmax.
- It may improve some RL algorithms.
- The proofs of (Property 4.1) and (Property 5.1) are simple and correct.
- The results are reproducible. I managed to run (partially) the experiment code provided as supplementary material.

Weaknesses:
- To my opinion the authors should integrate mellowmax policy, at least in a degraded form, to their tabular benchmarks (not only on HardSquare). The scipy implementation of the brentq algorithm could be accelerated by relaxing the default bounds and the  "xtol" and "rtol" parameters.
- Another solution is to learn the softmax temperature as in [Kim Konidaris 2019]. Did you try this as well ?
- For these reasons I am still wondering if the proposed method is a clear improvement against mellowmax and its tuned-softmax variants.


Doubts/Questions:
- I did not really catch the overemphasis argument on Figure 1a. According to the curves, a high-temperature softmax can place lower probability on a1 than a low-pressure/temperature resmax.
- The presented results and the lower-bound/gradient-bound proof machinery is straightforward and generic. It seems possible to generalize your results on a wider family of soft-greedy operators. What about a softmax where another convex function is used instead of exponential ? What about an ordinal "mean-of-top-k" operator ?

Minor remarks:
p2l-1 more greedy -> greedier
p3l26 b = \arg\max -> b \in \arg\max (or the uniqueness assumption should be set on the table)
p3l-7 "with results in a small environment" : ?? unclear to me
p5 Prop 4.1 give the softmax lower bound as well
p5 Prop 5.1 "temperature" or "pressure" as on page 3 ?
p6 proof : getting rid of the p and q on equation (7) and follow-up could save a few precious lines for your reproducibility statement
p6l-5 "show resmax" -> "show that resmax"
p7 Figure 3 "Stochastic RiverSwim" -> "Stochastic-reward RiverSwim" (it's only clarified in appendix)


**Summary Of The Paper:**

The most commonly used exploration policies for SARSA/Q-learning are epsilon-greedy and softmax sampling.
The problem with epsilon-greedy is that it explores obliviously the action space which may slow-down convergence of the Q function estimate, especially when the environment requires long sequences of actions to reach high rewards. On the other hand softmax provides a more informed exploration strategy, but as pointed out in [Assadi & Littman 2017] it is not a non-expansive operator and it is hence not guaranteed to converge toward a unique Bellman fixed-point.
This article propose a novel exploration policy called ResMax. The authors first show that ResMax is sufficiently exploring by exhibiting a simple lower bound on action selection probability (Property 4.1). They then show that the corresponding soft-greedy operator is non-expansive by bounding its gradient (Property 5.1).
Some experiments are then provided both with tabular and fitted Q function estimates. These experiments show empirically that ResMax is
outperforming both epsilon-greedy and softmax on small (but misleading) MDPs and Atari environments.
The most direct concurrent exploration policy: mellowmax proposed by [Assadi Littman 2017] is a variant of softmax where the temperature is adapted at each step. It is non-expensive but it requires to solve a linear root-finding problem a each step.
According to the authors this overhead was too costly to integrate mellowmax to their benchmarks. On the contrary, the ResMax policy is simple and quick to compute.


**Summary Of The Review:**

A novel non-expansive and simple exploration policy called ResMax.
I read this simple and direct paper with pleasure and it wakened a few questions in my head. This is a reason for me to support the paper.
On the other hand, due to the lack of experiments comparisons, I am still wondering if the proposed method is a clear improvement against mellowmax and the host of tuned-softmax variants that come to mind.

---

> ### Author Response · Authors · 2021-11-20
> **Thank you for your constructive review**
>
> We thank you for your constructive and detailed review of our work. This review makes really good suggestions for improving the paper and highlights the generalizability of our theoretical work to other soft-greedy operators.
> --- -
>
> ### Running mellowmax policy on the tabular experiments ###
> Thanks for making this suggestion. As per your suggestions, we ran the mellowmax operator with relaxed bounds and parameters. Results of tabular experiments on the riverswim and stochastic riverswim environments can be found here: https://anonymous.4open.science/r/resmax_211118-BD5F. Our results show that mellowmax does indeed have very good performance on these environments, but comes at the cost of a ~1000% increase in runtime.
>
> We would also like to remark that mellowmax was substantially more difficult to implement than other methods; finding appropriate values for the parameters of brentq took some trial and error. This fact, and the increased runtime of mellowmax, means mellowmax lacks the simplicity in implementation that softmax, resmax and epsilon-greedy share.
> --- -
> ###  Learning softmax temperature ###
> In the paper, we wanted to study these operators in their simplest form without using scheduling techniques. So, we haven’t tried this technique. However, since there is no code available for their technique, we could not manage to implement this technique in the short rebuttal time that we have.
> --- -
> ### Clarifying the overemphasis argument ###
> The overemphasis argument mainly points out that because of the usage of exponentials in the softmax function, it puts a lot of emphasis on the max action and this is not desirable in some environments that require more exploration.
>
> While it is true that one can find a sufficiently high value of temperature so that softmax assigns less probability than resmax with a low-valued \eta, in Figure 1a, we merely wanted to show how the emphasis on the greedy action will increase as we decrease the temperature for resmax and softmax. This is done to show that resmax’s emphasis on the greedy action will increase more slowly than softmax’s.
> --- -
> ### Generalizing the theoretical results on a wider family of soft-greedy operators ###
> We plan to include proofs for other soft greedy operators in the appendix of our paper using the “lower-bound/gradient-bound proof”. We would like to mention that this style of proof for showing non-expansion hasn’t been used before in showing non-expansion for soft-greedy operators in RL, most of the existing proofs are either proof by counter-example or operator specific. We hope this type of proof will significantly ease the future analysis in showing soft-greedy operators in RL are non-expansions.
> --- -
> ### Minor modifications to the paper ###
> These suggestions are really helpful. We are going to reflect these changes in the updated version of the paper.

---

### Official Review · Reviewer_sPif · 2021-11-01

**Correctness:** 3
**Technical Novelty And Significance:** 2
**Empirical Novelty And Significance:** 2
**Recommendation:** 3
**Confidence:** 4

**Main Review:**

Strengths :
	- The resmax operator achieves best of both worlds - an epislon greedy like operator and the softmax operator, which takes into account the optimality gap of value functions.
	- Figure 1 demonstrates the usefulness of the resmax operator on a simple MDP. It clearly shows how the resmax and mellowmax operator are considerably better than the softmax operator.
	- The experiments in the paper demonstrates the usefulness of the proposed operator.



Weakness :
	- The paper does not address very recent works that addresses convergence and optimization issues of the softmax operator. The usual drawbacks of the softmax operator, as discussed in the paper, are known but does not provide new insights to existing literature.
	- It is not clear to me how this operator influences or addresses the optimization issues (as mentioned in the related works below). During the early training stages, the optimality gap is rather poor (and in practice we do not have a good approximation to the optimality gap either). How does the approach take account of this? Considering the optimality gap between poor estimates of the value function can lead to sub-optimal convergence?
	- The paper claims "that the resmax operator is guaranteed to provide sufficient exploration" - I think simply justifying this based on visitation counts on simple problems is not enough. It is not clear to me how this operator balances trade-off between the exploration and optimization issues, even in simple RL problems.
	- The proposed operator is not theoretically justified. Proposition 5.1, even though is a good example - it is not clear to me why one should use the resmax operator in practice - since it seems that the claim is to propose an alternative to the softmax operator? It would have been helpful if the authors could at least comment on the convergence and optimality issues with the resmax operator, in a simple policy gradient setting? The softmax operator is often popular for PG learning - and recent works have established convergence rates to the global optima for softmax policies too. How does the resmax operator compare in contrast to the existing works supporting benefits and drawbacks of the softmax operator?
	- Experiments do not well justify why the proposed resmax operator may be helpful. It seems that there are several tasks where the beneftis are presented, but given existing works discussing issues with softmax, I am not surprised by the experimental results





Other comments :
	- The paper claims that the proposed resmax operator has better properties unlike softmax or epsilon greedy operators.  There are few lines of work similar in motivation, addressing issues of the softmax operator,  "Escaping the Gravitational Pull of Softmax" (Mei et al) and "Softmax Deep Double Deterministic PGs" (Pan et al) which addresses issues based on early sub-optimal convergence of the softmax operator.
	-  Can the authors discuss about the optimization issues, similar to Mei et al., for the proposed resmax operator? On those simple counter-examples, how does the resmax operator perform, and does it avoid the local optimas?
	- Perhaps not entire related : The RL literature often uses entropy regularization with softmax policies - and we know several benefits of the entropy regularizer (in terms of optimization landscapes, faster convergence rate etc). It seems to me that with the proposed resmax operator, which depends on value functions - the entropy regularization would not have the same effect? Can the authors comment on the use of entropy regularizers with resmax operator based action selection?


**Summary Of The Paper:**

This paper proposes an alternative greedy operator for RL which selects actions proportional to the optimality gap of value functions.  The claim is that it can explore better compared to an epsilon greedy approachand has similar benefits to a softmax operator. The main advantage of the proposed operator is that the probabiliteisof the actions do not converge too quickly, as in softmax operators.


**Summary Of The Review:**

I do not think this paper has enough novelty for acceptance. The proposed operator is indeed novel, but I can see several drawbacks of the operator, mainly since it can depend on very poorly estiamted value functions. Morever, the paper claims to provide an alternative to the softmax operator - there are quite some related works discusisng issues with softmax, and proposes alternative operators which can help with faster convergence, or solving the counter-examples of PG divergence with softmax policies. The softmax operator is well used in the policy gradient literature - and it is surprising that the paper completely ignores any issues or discusses anything about the effectiveness of the resmax operator in PG learning context.


The lack of discussions to related works (both theoretical and empirical) is a major red flag for me, since I do not know how reliable this resmax operator can be. Empirical experiments are not enough - and by that I do not mean large empirical studies showing effectinvess of resmax; but there are several counter-examples that exists in the literature (e.g from Mei et al) that discusses issues of the softmax operator. It would have been helpful to see why the resmax operator can be considered as an alternative to softmax (e.g like the Escort Transform proposed by Mei et al) in the context of those experiments.

---

> ### Author Response · Authors · 2021-11-20
> **Thank you**
>
> There seems to be a confusion about our problem setting. We are not looking at softmax for policy gradient methods. Rather, this is about putting the softmax operator around action-values. These are importantly different because in the case of policy gradient techniques we directly optimize action-preferences which are not the same as action-values. For example, parameterized softmax policies learned using policy gradient updates can concentrate and become (nearly) greedy. A softmax policy on action-values maintains non-negligible probability on each action, proportional to its value.
>
> --- -
>
> ### Addressing the related works in the policy gradient literature ###
> As noted, we are not in the policy gradient setting. For this reason we have not discussed the recent papers on the issues and advantages of softmax for policy gradient techniques such as "Escaping the Gravitational Pull of Softmax" (Mei et al) and "Softmax Deep Double Deterministic PGs" (Pan et al) in detail. We will add a comment in the paper clarifying this distinction.
> --- -
> ###  Convergence and optimality issues in policy gradient setting ###
> Since resmax is a non-expansion it is also Lipschitz continuous with a Lipschitz constant of 1. Thus we can invoke Theorem 1 of (Perkins and Precup, 2002) which states that the sequence of policies generated by approximate policy iteration using an epsilon-soft and Lipschitz continuous operator converges to a unique limiting policy regardless of the initial policy. Thus we also have convergence guarantees for policy iteration with resmax
>
> “Perkins, T.J., & Precup, D. (2002). A Convergent Form of Approximate Policy Iteration. NIPS.”.
> --- -
> ### The effects of using entropy regularizers with resmax ###
> Entropy regularization is used with policy gradient methods. The exploration parameter eta actually plays a somewhat similar role, in that it ensures we have a soft policy.

---

### Official Review · Reviewer_Y4xX · 2021-11-02

**Correctness:** 3
**Technical Novelty And Significance:** 3
**Empirical Novelty And Significance:** 2
**Recommendation:** 5
**Confidence:** 3

**Main Review:**

**Strengths**
* The proposed method is simple and widely applicable. Boltzmann policies are commonly used and improvements to their learning would affect many  This method has only one hyperparameter and requires no extra compute.
* The proposed operator has nice mathematical properties and seems well-behaved (unlike softmax).
* The paper brings its theory all the way through to deep learning experiments.
* The writing is mostly very clear.

**Weaknesses**
* The various hyperparameter comparisons between different methods are somewhat confusing, and I worry that they could be misleading. For example, in Figure 3 the hyperparameters for softmax, resmax, and $\varepsilon$-greedy are put on the same axis. It's hard to draw conclusions from this, given that e.g. $\varepsilon$ lives on a different scale than $\tau$, and $\tau$ and $\eta$ are not obviously comparable either. The results in the bar charts in Figure 5 will also depend strongly on the authors' choices for hyperparameters corresponding to varying amounts of exploration for each method.
* The title of Section 6.1 claims that resmax leads to more exploration than softmax, but I don't see any direct evidence for this. On some environments resmax leads to better performance, and it seems potentially less sensitive to its hyperparameter (although maybe the hyperparameters just live on different scales), but there is no clear empirical analysis of the amount of exploration, e.g. state coverage.
* The crucial question for a method like this is whether or not it makes a difference overall when used more widely. It would make the case a lot stronger to have a broader set of experiments with deep learning, though I understand computational resources might be a limiting factor.
* The work could use more analysis (theory or experiment) of convergence properties, along the lines of the "softmax gravity well" paper.


**Summary Of The Paper:**

This paper introduces a new soft operator, resmax, for mapping Q-values to action probabilities. This operator is designed to replace softmax in Boltzmann-style policies while having the non-expansion property which enables the convergence of Q learning.  The paper provides theory demonstrating the coverage and non-expansion properties of resmax, as well as somewhat more heuristic evidence that resmax enables more exploration than softmax.

**Summary Of The Review:**

The new resmax operator proposed in this work seems interesting and could be a more robust choice for Boltzmann-type policies. The paper contains straightforward reasoning supporting its candidacy, but falls short in really proving the claims that resmax outperforms softmax in terms of (1) exploration, (2) convergence to the optimal policy, and (3) deep learning practice. Nonetheless the method seems interesting and worth trying more widely.


### After responses

After reading the other reviews and considering the authors' responses I am leaning towards rejecting this paper. The results are potentially interesting but not compelling, and I do not find that the analysis in this work significantly illuminates the problem otherwise.

---

> ### Author Response · Authors · 2021-11-20
> **Thanks for the important questions**
>
> We thank the reviewer for bringing up these important questions. This review points out the parts of the paper that require further clarification and asks for a more detailed empirical study.
> --- -
>
> ### Selecting fair values of exploration parameters for our experiments ###
> Thanks for bringing up this important point. To make a fair comparison between the operators, we swept over a large set of hyperparameters for each soft-greedy operator to make sure that the exploration parameter with the near-best performance resides in this set.
>
>
> Our experiments in the atari environments showed that all the best exploration parameters for softmax and resmax resides somewhere between  $2^{-24}$ and $1$. However, the exploration parameters for softmax and resmax with the same value results in different level of exploration. To select a fair value for low, medium, and high exploration parameter in atari environments, we tried a large set of different values of exploration parameters in the range  $2^{-24}$ to $1$ across Asterix and Pong as well as our sensitivity plots results from simple environments like MountainCar and Lunar Lander. Our results indicated that Softmax majorly achieves a good performance with three values of temperature $2^{-16}$, $2^{-8}$ and $2^{0}$ and resmax achieves a good performance with $2^{-24}$, $2^{-16}$ and $2^{-8}$. Therefore, we used these parameters as the default for low, medium, and high exploration parameters across all the experiments.
>
>
> For $\varepsilon$-greedy on the other hand, we noticed that $\varepsilon=0.1$ always results in a near-best performance, so we only tried several values around it to see how increasing exploration changes its performance. Based on this, we have chosen the exploration parameters for each operator and presented its hyperparameter sensitivity plots in the appendix.
> --- -
> ###  A Different Visualization of Figure 3 ###
> We completely agree that the Figure 3 is hard to analyse because of the way that we visualize the hyperparameters. So, we come up with a better visualization that shows the results of the operator in the same scale.
> We have included an updated version of Figure 3 with each hyperparameter max-min scaled here: https://anonymous.4open.science/r/resmax_211118_2-BFD8/.
> --- -
> ### Does ResMax lead to more Exploration than softmax? ###
> The only way resmax can reach higher performance than other soft-greedy operators on the hard exploration problems like RiverSwim is to promote more exploration. In RiverSwim, the agent has to pass all the states to reach the most rewarding state and increase its return. Therefore, it has to explore all the states sufficiently so it would be able to come up with a good policy, so a good level of state coverage is necessary for reaching high return (i.e., coming up with a good policy).
> --- -
> ### Extending the Experimental Results ###
> We believe the current set of results already provides a thorough characterization of the methods, but of course understand the desire for more results to validate claims. We ran our experiments on 5 other atari environments, including Pong, SpaceInvaders, Gravitar, Pitfall, and Venture. Gravitar, Pitfall, and Venture are hard exploration environments according to “On Bonus Based Exploration Methods In The Arcade Learning Environment” paper. Our experiments show that resmax consistently achieves similar or highly better performance than softmax and epsilon-greedy operators. Relevant detail is that we used the same setting as described in the paper for running these experiments, so we are only using these operators for decision making (i.e., exploration). The results can be found in this anonymous GitHub repository:
> https://anonymous.4open.science/r/resmax_211117-AC08/
> --- -
> ### Extending our Analysis to the Policy Gradient literature ###
> Our focus here is on soft-greedy operators for action-values. This is distinct from policy gradient methods. Softmax parameterizations for policy gradients have much different behavior, because the action preferences can be learned to concentrate much more aggressively. When putting soft operators on action-values, the action-values themselves are constrained to be an estimate of expected returns. The "softmax gravity well" paper focuses only on policy gradient techniques, so it would be out of the scope of our paper thesis to develop theories or experiments in the direction of this paper.

---

### Official Review · Reviewer_NLrw · 2021-11-02

**Correctness:** 3
**Technical Novelty And Significance:** 3
**Empirical Novelty And Significance:** 2
**Recommendation:** 6
**Confidence:** 5

**Main Review:**

There is always a trade-off in reinforcement learning between taking actions that are estimated to be high-rewarding, and trying the less rewarding actions to learn if better options are available. A standard technique to addressing this trade-off in large-scale reinforcement learning is to use operators/functions that assign high probability to the best action, but also put some non-zero probability behind the less rewarding ones. While this approach is generally not provably sample efficient, it is often easy to apply and very competitive. Examples include epsilon-greedy, Boltzmann, and mellowmax.

I find it useful to make a distinction between the operator used in value-function optimization (such as softmax), and the policy used for decision making (such as soft argmax). The current draft sometimes recognizes the distinction, but sometimes conflates the two by generically referring to both as operators.

On this very note, it is not that mellowmax "requires" solving for a root finding algorithm. If the objective is to have an on-policy algorithm, i.e one that uses the same form in value-function optimization and decision making, the way to obtain that is by solving the root-finding algorithms. In fact, previous work has shown the impact of using Boltzmann/mellowmax just for value function optimization and not in decision making (See Song and others "Revisiting the softmax bellman operator: New benefits and new perspective"). So parts of the paper need revision to clearly state this point.

In the background section, stating that q* is the value for "the" soft optimal policy is incorrect. There may generally be multiple optimal policies, though they all yield "the" same q*.

Going to the main contribution, namely resmax, it is interesting to note that the behavior of resmax can be qualitatively different than epsilon greedy when more than one greedy actions co-exist. In particular, epsilon greedy will deterministically chose an action (assuming we are breaking ties non-randomly) but resmax would yield equiprobable policy. On this note, I was wondering if authors have any thoughts about generalizing this to the continuous control setting where the notion of action gap may be difficult/impossible to compute.

The paper alleges that softmax overemphasizes good actions, and it is desirable to reduce this overemphasis. While this is intuitively plausible, overemphasis does not sound like an objective thing to me. Why is the amount of emphasis put by resmax the right one? why not to emphasize the gap even less than resmax does? Is the least emphasizing operator, namely uniformly at random, the best choice? well clearly not, so what’s the right strategy if one exists at all? More about this on my note about the proof of the non-expansion.

Property 4.1 seems too trivial to me to warrant being in the main text.

I checked the proof of Property 5.1 and it is correct. That said, i feel like the proof could have been generalized with some moderate work. In particular, right now probabilities are proportional to 1/delta where delta could be the gap. But in general this could have been 1/f(delta) where f is any positive and non-decreasing function. In that sense one does not have to commit to the current (identity) f but in general can have square root, quadratic, or even exponential f. I believe it is possible to generalize the proof to arbitrary f, which in turn also addresses my concern about the arbitrary amount of emphasis obtained by using f=identity.

The paper provides experimental results on 3 toy domains as well as 3 Atari domains. On the toy domains, I see two odd choices: 1- the mountain car task has been altered. I generally prefer not to see any modification to standard benchmarks as this makes it impossible to cross-check performance with other existing results 2- there is a strange focus on the size of the replay buffer and the robustness of different algorithms with respect to it. I can't see why this is interesting to study/relevant to the thesis of the paper.

Finally, for Atari tasks, it is unclear to me why these three domains are chosen. It would have been meaningful to only choose domains that are hard for exploration. To be fair, Freeway is one such domain, but it also lends itself to the shockingly simple policy of always moving up.



**Summary Of The Paper:**

The paper introduces a new operator for exploration in reinforcement learning. The new operator, named resmax, is shown to be a non-expansion, therefore ensuring convergence in learning and planning. Resmax is also competitive in practice on a set of benchmarks tested.

**Summary Of The Review:**

Overall, it is useful to have an additional operator/policy that is somewhat similar to softmax but with its own strengths and weaknesses. The form of the operator is a little ad hoc and too specific, but all things considered I lean towards acceptance here.

---

> ### Author Response · Authors · 2021-11-20
> **Thanks for your detailed and constructive review**
>
> We thank you for your constructive and detailed review of our work. This review makes useful suggestions to improve the paper.
> --- -
>
> ### The distinction between soft-greedification and exploration ###
> We did try to be clear about the two roles, but can see how naming obfuscated that attempt. What we will do is introduce new terminology: the resmax policy and the resmax operator. The resmax policy is the first definition given, defining the soft-greedy policy on the action-values, and the resmax operator is the corresponding operator used in the Bellman update.
>
>
> It is true that if we use mellowmax for value-function optimization and not decision making, there is no need to solve the root-finding problem. However, in this paper we are focused on using operators for exploration (i.e., decision making), and so always use soft-greedy operators for decision making throughout our experiments. We will clarify that mellowmax only has this limitation when the policy needs to be queried for decision-making. We will further discuss that these operators can be useful just in the update, even if they are not used for exploration, and cite the results from Song et al.
> --- -
>
> ### Resmax for continuous control ###
> One possible direction is to use resmax to avoid overestimation in the critic in continuous action control with DDPG. As you mentioned in the Song et al. paper, it can be beneficial to use this operator just in the value function update, even if we then use a greedy policy. DDPG attempts to learn the greedy policy, but the values could be updated using a soft-greedy policy. The action outputted by the greedy policy could provide an estimate of the greedy action, which we could then use in the suboptimality gap in resmax. In the value function update, we would need to approximate the integral over actions or find an efficient action sampling approach. A recent paper (“Softmax deep double deterministic policy gradients”) has one strategy to do so; pursuing this is a natural next step.
> --- -
> ### Clarifying the overemphasis property ###
> We argue in the paper that softmax puts more emphasis on the greedy action than resmax, and that this may not be desirable in many environments. It is still true that this overemphasis of softmax on the greedy action can actually improve the results in some environments, especially the class of environments where a nearly greedy policy is effective and the agent does not need to explore much. Though we as yet do not concretely know when overemphasis will hurt or help, our experiments show that across several environments the overemphasis from softmax results in worse performance than resmax.
> --- -
> ### Transferring Property 4.1 to the appendix ###
> As the reviewer pointed out “Property 4.1 seems too trivial to me to warrant being in the main text.” Therefore, we will transfer it to the appendix in the updated version of the paper.
> --- -
>
> ### Reasons for modifications to the MountainCar environment ###
>
> Thanks for pointing this out. Sparse MountainCar environment is a standard environment that people already are using for benchmarking the exploration algorithms. For instance, “Temporally-Extended $\epsilon$-Greedy Exploration” and “Context-Dependent Upper-Confidence Bounds for Directed Exploration” use this environment to benchmark their exploration techniques. In the paper, we called it MountainCar-v0 from the OpenAI-gym and this resulted in this confusion. Therefore, we are going to clarify this by calling it Sparse Mountain Car environment in the updated version of the paper.
> --- -
> ### Extending results to more hard exploration environments  ###
> Thanks for bringing up this important question. We chose these three environments to show that this operator can work in hard exploration atari environments in addition to atari environments that are normally used and do not need a high level of exploration. Prior to the submission time, we could not run them on more hard exploration environments because of computational constraints. However, since then we ran our experiments on 5 other atari environments afterward, including Pong, SpaceInvaders, Gravitar, Pitfall, and Venture. Gravitar, Pitfall, and Venture are hard exploration environments according to “On Bonus Based Exploration Methods In The Arcade Learning Environment” paper. Our experiments show that resmax consistently achieves similar or highly better performance than softmax and epsilon-greedy operators. Relevant detail is that we used the same setting as described in the paper for running these experiments, so we are only using these operators for decision making (i.e., exploration). The results can be found in this anonymous GitHub repository:
> https://anonymous.4open.science/r/resmax_211117-AC08/

---

> > ### Author Response · Authors · 2021-11-20
> > **Additional Comments**
> >
> > ### Empirical Study on the size of the Replay Buffer ###
> > The main thesis of the paper is using resmax for exploration and in this section, we are hypothesizing that since resmax results in better exploration, it fills the replay buffer with more useful transitions. This results in resmax being more robust to the change of the replay buffer.
> > --- -
> > ###  Clarifying our claim about q* ###
> > We agree that it is confusing to use the same notation for the optimal policy and soft optimal policy. So, as you pointed out, we will correct this part of the paper by saying that q* is the value for the optimal policy and not the soft optimal policy.
> > --- -
> > ### Generazability of Property 5.1 ###
> > Our current understanding is that certain gap emphasis functions like $f(\delta_i) = e^{\delta_i}$ where $\delta_i = x_j - x_i$, resmax can no longer a non-expansion. To clarify, for gap emphasis function $f(\delta_i) = e^{\delta_i}$ if $x= [56,53], y = [81, 67]$, and $\eta= 50$ then $|rm(x,eta) - rm(y,eta) > ||x-y||_\infty$, since $|56 - 81 - 1.24| > |56-81|$. Also with gap emphasis function $f(\delta_i) = \delta_i^p$ for $p>1$ resmax is no longer a non-expansion see the proof below.
> >
> > Thus our selecting probabilities proportional to $1/\delta$ seem to be necessary in order to maintain resmax's non-expansion property.
> >
> > We can also do a similar proof to find when values of $p$ would work for the gap emphasis function $f(\delta_i) = p^{\delta_i}$ if the reviewer would like to see this.
> > ### Empirical Study on the size of the Replay Buffer ###
> > The main thesis of the paper is using resmax for exploration and in this section, we are hypothesizing that since resmax results in better exploration, it fills the replay buffer with more useful transitions. This results in resmax being more robust to the change of the replay buffer.
> > --- -
> > ###  Clarifying our claim about q* ###
> > We agree that it is confusing to use the same notation for the optimal policy and soft optimal policy. So, as you pointed out, we will correct this part of the paper by saying that q* is the value for the optimal policy and not the soft optimal policy.
> > --- -
> > ### Generazability of Property 5.1 ###
> > Our current understanding is that certain gap emphasis functions like $f(\delta_i) = e^{\delta_i}$ where $\delta_i = x_j - x_i$, resmax can no longer a non-expansion. To clarify, for gap emphasis function $f(\delta_i) = e^{\delta_i}$ if $x= [56,53], y = [81, 67]$, and $\eta= 50$ then $|rm(x,eta) - rm(y,eta) > ||x-y||_\infty$, since $|56 - 81 - 1.24| > |56-81|$. Also with gap emphasis function $f(\delta_i) = \delta_i^p$ for $p>1$ resmax is no longer a non-expansion see the proof below.
> >
> > Thus our selecting probabilities proportional to $1/\delta$ seem to be necessary in order to maintain resmax's non-expansion property.
> >
> > We can also do a similar proof to find when values of $p$ would work for the gap emphasis function $f(\delta_i) = p^{\delta_i}$ if the reviewer would like to see this.

---

> > > ### Author Response · Authors · 2021-11-24
> > > **If $p>1$ then resmax with gap emphasis function $f(\delta) = \delta^p$ is no longer a non-expansion.**
> > >
> > > The Resmax operator with gap-emphasis function $g(v) = v^p$ for $p \in \mathbb{R}^+$ is defined as follows
> > >
> > > $
> > >     \text{rm}(\vec{x},\eta) = \sum_{i \in [d], i \neq j} \frac{x_i}{d + \frac{1}{\eta}(x_j - x_i)^p} + x_j \left[1 - \sum_{i \in [d], i\neq j} \frac{1}{d + \frac{1}{\eta}(x_j-x_i)^p}\right] = \sum_{i \in [d], i \neq j} \frac{x_i - x_j}{d + \frac{1}{\eta}(x_j - x_i)^p} + x_j
> > > $
> > >
> > > where $j \doteq \arg\max_i x_i$. To show Resmax is a non-expansion we will need the following definition which states for Lipschitz function $f(x)$, $f: \mathbb{R}^d \rightarrow \mathbb{R}$,
> > >
> > > $
> > >     \left \lvert f(x) - f(y) \right \rvert \leq L_p \lVert x - y \rVert_q
> > > $
> > >
> > > where $L_p = \sup \{\lVert \nabla f(x) \rVert_p : x \in D\}$ is the Lipschitz constant, $D$ is a compact set, $\nabla f(x) = (\partial f/ \partial x_1,...,\partial f /\partial x_d)$ is the gradient of the function $f(x)$, and $1/p + 1/q = 1, 1 \leq p,q \leq \infty$. Now we want to show that if $p >1$ resmax with gap emphasis function $g(v) = v^p$ is no longer a non-expansion. Let $d = 2, \eta = 1, j = 2,$ and $\delta_1 \doteq x_2 - x_1 \geq 0$. We take the derivative of resmax with respect to $x_1$ and get
> > >
> > > $
> > >     \frac{\partial }{\partial x_1}\text{rm}(\vec{x},\eta=1) =   \frac{-p\delta_1^p + \delta_1^p + 2}{(2 + \delta_1^p)^2} = \frac{2 + (1-p)\delta_1^p}{(2 + \delta_1^p)^2}.
> > > $
> > >
> > > Now we compute $\partial \text{rm}(\vec{x},\eta=1)/ \partial x_2$,
> > >
> > > $
> > >     \frac{\partial }{\partial x_2}\text{rm}(\vec{x},\eta=1)  =  \frac{(p-1)\delta_1^p - 2}{(2 + \delta_1^p)^2} + 1 = \frac{(p+3)\delta_1^p + \delta_1^{2p} + 2}{(2 + \delta_1^p)^2}.
> > > $
> > >
> > > Note that $\partial \text{rm}(\vec{x},\eta=1)/ \partial x_2 \geq 0$ when $\delta_1 \geq 0$ and $p\geq-3$ we will use this fact later. Since we have $\nabla \text{rm}(\vec{x},\eta=1)$, we will define and compute $L_1(x)$,
> > >
> > > $
> > >     L_1(\delta_1) \doteq \left  \lvert  \frac{\partial }{\partial x_1}\text{rm}(\vec{x},\eta=1) \right \rvert + \left \lvert \frac{\partial }{\partial x_2}\text{rm}(\vec{x},\eta=1)\right \rvert = \left  \lvert \frac{2 + (1-p)\delta_1^p}{(2 + \delta_1^p)^2} \right \rvert + \left \lvert \frac{(p-1)\delta_1^p - 2}{(2 + \delta_1^p)^2} + 1\right \rvert.
> > > $
> > >
> > > We would like to note that if $p=1$ then $L_1(\delta_1) = 1$ for all $\delta_1 \geq 0$. Now we will show that if $p > 1$ then $L_1(\delta_1) > 1$ for some  $\delta_1$. Since all the terms of $\frac{\partial }{\partial x_2}\text{rm}(\vec{x},\eta=1)$ are positive, we can remove the absolute value from this term.
> > >
> > > Now let $\delta_1 = (2c/(p-1))^{1/p}$ for $c \in [1,\infty)$, which when $p>1$ is positive, then we have
> > >
> > > $
> > >      L_1(\delta_1)  = \left  \lvert \frac{2 + (1-p)(2c/(p-1))}{(2 + 2c/(p-1))^2} \right \rvert +  \frac{(p-1)(2c/(p-1)) - 2}{(2 + 2c/(p-1))^2} + 1\\
> > >      = \frac{(p-1)(2c/(p-1)) - 2 }{(2 + 2c/(p-1))^2} +  \frac{(p-1)(2c/(p-1)) - 2}{(2 + 2c/(p-1))^2} + 1 = \frac{4c - 4}{(2 + 2c/(p-1))^2} + 1.
> > > $
> > >
> > > Now for $c > 1$ we have that
> > >
> > > $
> > >     L_1 \geq L_1(\delta_1) =  \frac{4c - 4}{(2 + 2c/(p-1))^2} + 1> 1
> > > $
> > >
> > > where the first inequality holds by the definitions of $L_1$ and $L_1(\delta_1)$ and the second strict inequality holds since $4c - 4 > 0$ when $c>1$.
> > > Putting this all together, we have shown that if $p > 1$ for the gap emphasis function $g(v) = v^p$ and $\eta = 1$ then there exists a vector $\vec{x} \in \mathbb{R}^2$ such that if $\delta_1 \doteq x_2-x_1 = (2c/(p-1))^{1/p}$ for $c \in (1,\infty)$, then $L_1 > 1$. This means that if we put more emphasis on the gaps, resmax is no longer a non-expansion.
> > >
> > > This proof can be generalized for arbitrary $d$ by letting $j= \arg\max_i x_i = d$ and considering the special case when $\delta_1=\delta_2=...=\delta_{d-1} \approx (cd/(p-1))^{1/p}$. Thus the current gap emphasis function for resmax is tight in the sense that adding more emphasis to the gaps would mean resmax is no longer a non-expansion.

---

### Decision · Program_Chairs · 2022-01-20

**Decision:**

Reject

**Comment:**

This paper proposes a new softmax like operator, to be used instead of eps-greedy or softmax in Q learning algorithms. There has been some previous work in this direction, most notably Mellowmax, but the proposed operator is more computationally efficient, and there is some experimental evidence that it improves DQN performance.
The reviews were mixed, with two mildly positive reviewers (6), who found the work interesting, and two negative reviewers (3,5), who raised issues about the impact of the work when taken as a part of a larger RL algorithm, and about the generality of the work w.r.t. to other RL algorithms like policy gradients. During the discussion, the reviewers did not reach an agreement.
My decision to reject the paper is based on the following: while the idea is novel, and the contraction analysis is appropriate, the main interest to the community in such an idea is either experimental - can it be used to push the state of the art RL algorithms? or theoretical - can we glean new theoretical insights using this method? In its current presentation, there is not enough evidence in the paper to support either of these.
I encourage the authors to either dig deeper into the experimental evaluation and produce more convincing results, or dig deeper into the theory and show some theoretical benefit of Resmax.